# Building an Augmented Reality Experience on Top of a Smart Pavement Management System

Maryam Moradi *  and Gabriel J. Assaf

Faculty of Construction Engineering, École de Technologie Supérieure, 1100 Notre-Dame St. W.,
Montreal, QC H3C 1K3, Canada
* Correspondence: maryam.moradi.1@ens.etsmtl.ca

**Abstract:** Pavement Management Systems (PMS) offers a systematic collection, storage, analysis, and modeling of road condition data to optimize resources across a road network. Adding artificial intelligence (AI) and augmented reality (AR) to PMS could improve their technical or visual aspects. This paper tries to identify a method to improve the understanding of the consequences of the city council's decisions in the urban pavement management system field. This paper establishes the potential of AR. It provides future maintenance and rehabilitation (M&R) actions needed based on the recommendation of the future distress in the study area. The road cracks are discovered through technical analysis, and a CityEngine model is established based on the PMS results. Additionally, in terms of visualization, this paper's unique feature delivers the result as an AR experience. Applying the Unity game engine and importing the built CityEngine model and the embedded textures as input empowered us to provide a dynamic product in terms of data and analysis and a real-time Decision Support System (DSS) for the final users. This paper concludes that researchers need many different modules to design and implement an efficient PMS to move toward a smart PMS. The smart city concept is meaningless without a tight collaboration between all distinctive parts of each urban infrastructure management system. Additionally, this paper attempts to provide answers for researchers and an outlook for future research, the development of the proposed method, and its application in other fields

**Keywords:** intelligent pavement management; PMS; urban infrastructure; AR

## 1. Introduction

Roads connect more people and places, enabling economic growth and social development. Since the condition of road infrastructure impacts the citizen's quality of life [1], every action should be carefully planned because this is a highly complex and sensitive issue. During the planning process, municipal governments, as well as all the road administrations, face significant challenges. As part of metropolitan plans, road infrastructure design and administration are crucial since construction, maintenance, and corrective action budgets must be identified yearly [2].

Road maintenance in cities was reactive in the past, focusing on repairing what was already in a bad state. The modern method is proactive and focused on maintaining a goal level of performance over time by protecting pavement assets that are still in good shape. According to the proactive strategy, pavement management is an organized method of preserving, enhancing, and running a network of pavements [3]. The AASHTO (1993) defined the pavement management system (PMS) as "a set of tools or methods that assist decision makers in finding the optimum strategies for providing, evaluating, and maintaining pavements in a serviceable condition over some time" [4].

The PMS can determine the right time for maintenance activities based on pavement distress indices and predictions. Moreover, pavement performance can be compared by pavement type, traffic volume, or other attributes. This information allows preventive

maintenance programs to be developed, extending the life of pavements. Furthermore, it can simulate the budget requirements for maintaining pavement at different levels. However, there are still lots of obstacles to acquiring a smart PMS.

PMS is based on long-established monitoring methods that heavily rely on accumulated experience. In addition to updating PMS data collection methodologies and data analysis tools, technological advances like artificial intelligence (AI) and the Internet of Things (IoT) can be integrated into PMS decision support systems. With the help of intelligent technologies, we will be able to monitor and assess pavement conditions in a more effective, less expensive, safer, and environmentally friendly manner [5]. Due to the geospatial nature of every single element of a PMS, it is necessary to integrate the PMS and its visualization output with geographical information systems (GIS).

Developing a linear route system through GIS integration can enhance any PMS with pavement data. GIS can match the geographical characteristics of road networks using spatial analysis and is increasingly being integrated into PMS systems. The GIS-based PMS offers several advantages, including editing database queries, visualizing stats, and editing the database. Non-technical people will find it easier to understand a color-coded interactive 3D result. Answering the main research question, "What is the role of visualization technologies in the management of urban road infrastructure?" this study seeks to identify a method for improving the understanding of the consequences of city council decisions in a PMS based on an interactive visualization platform.

In 2016, when the industrial revolution shifted into a new phase [6], augmented reality (AR) emerged as the most important technical evolution. AR adds interactive virtual objects and pictures to real-world settings. Due to the fact that AR is interactive and transparent, it offers a technological advantage over virtual reality (VR). In addition, AR in PMS can help avoid costly mistakes, increase efficiency, persuade municipal officials to make more efficient investments in urban infrastructure, and save money [7]. To move toward a smart city, including a brilliant PMS, game engines are being applied to build augmented reality experiences. As an illustration, Figure 1 shows a simplified view of the AR experience in an urban area.

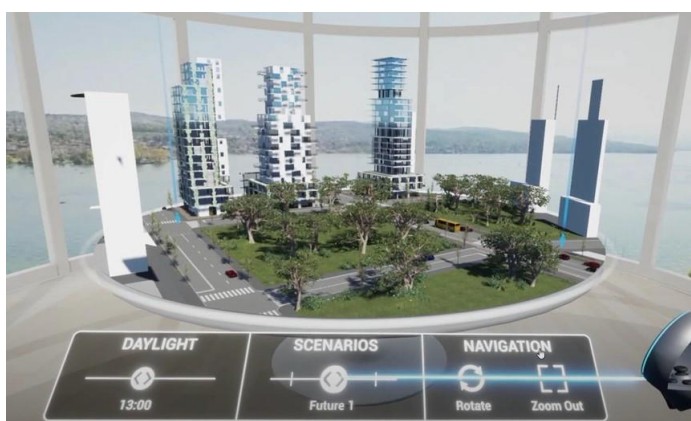

**Figure 1.** Schematic view of a city with AR experience, reprinted with permission from [8].

To achieve this research's primary goal, based on AI and AR's capabilities, an intelligent pavement management system is elaborated in this paper, and the results are presented. Gathering and analyzing massive amounts of data about urban road infrastructure maintenance is necessary for monitoring and maintaining them. A GIS would be perfect for helping us organize this data because it is spatial. Ground inspections of roads are expensive and require human resources. As a result, it is needed to increase efficiency, which helps manage and maintain road infrastructure and avoid wasting millions of dollars annually because of poor road infrastructure maintenance actions. The best option is to integrate AR technologies into a GIS framework to provide the best M&R actions for every PMS framework. AR and GIS integration gives up plenty of possibilities for more

intelligent infrastructure maintenance management applications. It is essential to reveal the hidden core knowledge of the infrastructure's component elements to properly understand user tasks and provide better feedback to the system and users. Information hidden in a component can include its cost, the history of its maintenance, and settings. This paper's contribution is proven by revealing the best possible M&R action hidden behind every road section in a PMS.

This article is divided into five major sections: Section 2 provides a broad review of the literature, including PMS, AR, GIS, and their interrelationship. Section 3 consists of a methodology followed in this study to meet the research objective and fill out the research gap. Section 4 introduces the study area, defines the data used in this research, and the produced result is presented. Then, this section reviews the validation procedure applied in this paper. Finally, Section 5 concludes the article and proposes future trends in the smart PMS and AR field.

## 2. Literature Review

Due to the aging of pavements and degradation, which is amplified by the growing demand for infrastructure and growing traffic loads [9], structured pavement management has become essential. Using GIS to improve pavement maintenance systems has been around for several years in various versions. University academics have used this strategy to study a few roads on campus [10,11] or in partnership with municipal roadway agencies to establish a pavement management system for urban regions [10–13]. In North Carolina and Arizona, comparable systems were developed in collaboration with universities and implemented in some districts [14,15]. There are several GIS platforms that researchers work with to make their readers ensure to what extent the produced results are accurate and reliable.

Several GIS software solutions for the design of such systems are mentioned in the literature, including ArcView [16,17], ArcInfo [12], MapInfo [14], ArcGIS [18,19], Geomedia Pro [20], and ESRI MapObjects [21,22]. These analyses can range in sophistication and plan for maintenance activities in the future. In addition to the GIS software, which mainly focuses on geospatial data analysis, some PMS applications emphasize the technical aspects, like providing the optimum M&R solution for each road section.

Previously, pavement management solutions like HDM-III and PAVER are proposed [23]. They depended on cost-benefit analysis and could not balance competing for asset types and traffic modes [24]. For asset management, most of these problems were solved by applying linear programming and other heuristic optimization methodologies [25,26]. Route and connection options are influenced by traffic perception, road condition, route capacity, accessibility, cash benefits, physical security, and to a lesser extent, environmental responsibility. Adding more models to the PMS improves its conceptual management and coherence with most economic expansion frameworks and makes it more attractive to new users [27]. The most comprehensive PMS includes a road dataset, a pavement condition evaluation method, and a decision-making device for maintenance costs and available resources [12].

Before starting the PMS implementation process, city agencies should assess the advantages, costs, and resources required. An assessment is needed for informed decision-making and public acceptance. Before alerting the public and elected officials about the benefits of PMS, each department must convey a cost-benefit analysis [28,29]. A PMS implementation's cost includes the initial setup and ongoing maintenance. The following are some of the factors that influence the cost of a PMS [29]:

- Collection of data and database creation,
- Obtaining and installing essential software,
- Staff training and consultant activities,
- Costs of pavement M&R.

Since quantifying all advantages and disadvantages may be challenging, an agency might do a fast study to determine the cost-effectiveness of deploying a PMS [29]. In the

past, surface distress assessment data were collected either by walking along the shoulder (walking surveys) or by driving alongside its shoulder (riding surveys, also called windshield inspections) [30,31]. These approaches take a long time, require costly equipment, and may create traffic issues. Nevertheless, data are automatically collected, even for surface distresses [32,33].

Data collection methods, as well as data processing methodologies, have an impact on the decisions made by these systems. Unmanned aerial vehicles (UAVs) [34], cars for research (essentially cars embedded with sensors) [35], cellphones [36], sensors inserted in the pavement [37–39], and electric vehicles [40] have all been studied by academics as innovative data collection methods for pavement monitoring.

On the other hand, a few pieces of research fill the gap between innovative data collection technologies and data analysis methodologies that are likewise based on novel intelligent approaches and suited for inclusion in DSS for smart pavements. Machine learning and other analysis technologies can extract meaningful information from this data, enabling pavement diagnosis and predictions. In addition, for a decision-making system, it is essential to have a relationship between the data and analysis section.

The two recent studies have demonstrated the utility of linking data acquisition and analysis approaches as a decision-making system for pavement management [41,42]. However, their current evaluation process is primarily concerned with examining the pavement's operational state (i.e., surface distress identification and prediction or roughness indicator estimation), ignoring the possibility of a more complex DSS concept that includes developed tools for structural and functional pavement monitoring [5].

Figure 2 depicts a DSS covering smart cities' data collection mechanisms and functional and structural pavement metrics analysis capabilities. This diagram also shows how the two primary components of a DSS work together in an intelligent PMS: Methods of data collecting, as well as data analysis and decision-making tools [5]. Smart PMS are unlikely to succeed without citizens sharing information about events in the city, despite the existence of automatic systems for collecting data and making analyses. In addition to enabling users (including either city administration or citizens) to manipulate virtual objects in natural environments, AR can facilitate road management with significantly less time and expense.

The fourth industrial revolution has increased the implementation of AR in urban development studies. Figure 3 illustrates the number of research articles published, indicating that interest in the AR field has grown since the fourth industrial revolution. AR has seen significant growth and implementation in civil infrastructure in recent years compared to its early years [43–46].

Based on the capabilities of game engines in producing AR experiences [47], there are two main methods for AR applications categorization in civil infrastructure: by item (bridges, large structures, and rail lines) or by real applications (engineering layout, Structural Health Monitoring (SHM), structural design, manufacturing, and others [48]. Figure 4 illustrates the proportion of AR applications in engineering structures from 2016 to 2020, broken into five categories: smart city, construction, building information modeling (BIM), SHM and damage detection, and subsurface utilities [49].

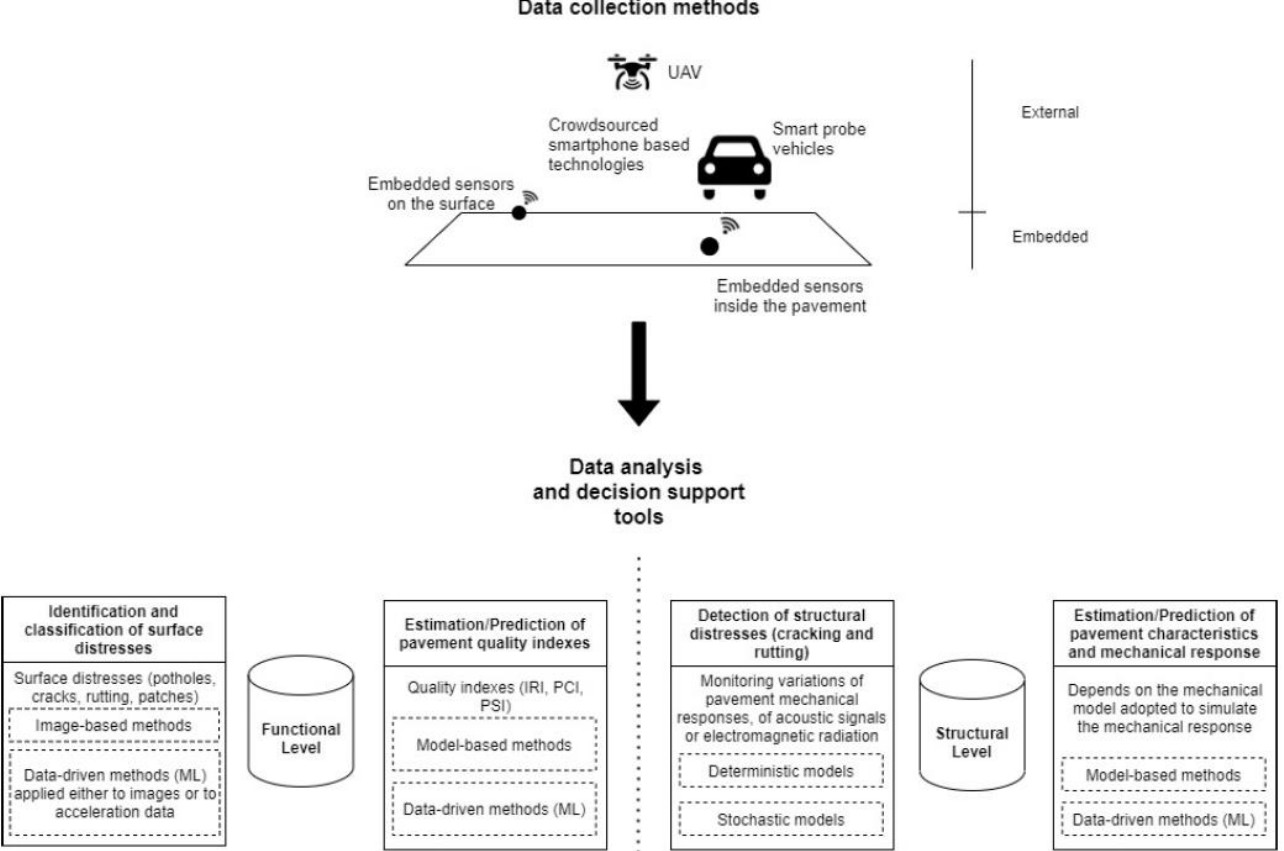

**Figure 2.** A decision support system for intelligent pavement, reprinted with permission from [5].

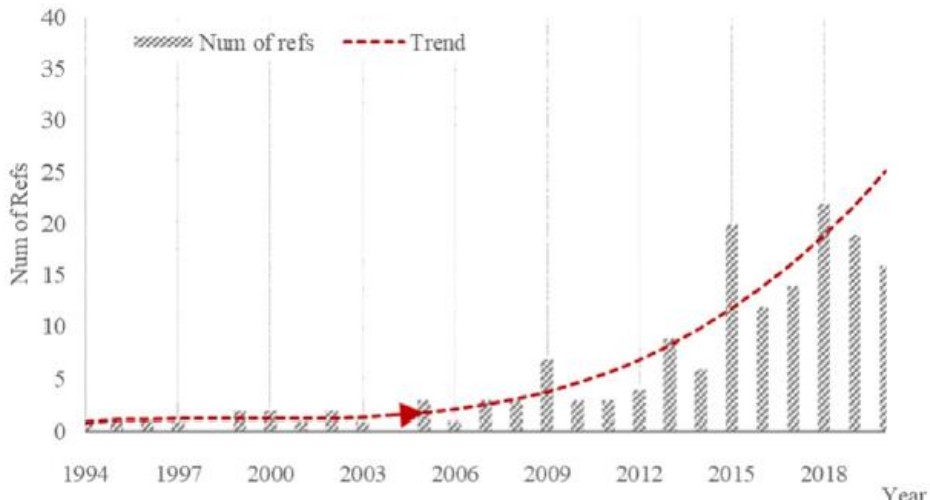

**Figure 3.** Civil infrastructure in AR deployment growth, reprinted with permission from [7].

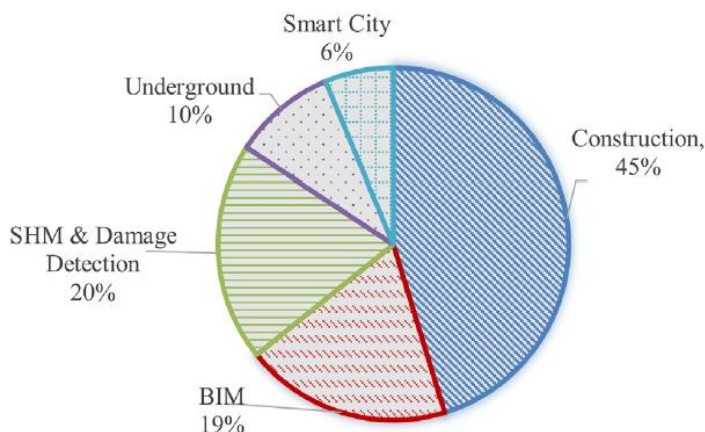

**Figure 4.** AR used in engineering structures as a percent (2016–2021), reprinted with permission from [7].

Researchers should consider every specification when deciding on an appropriate AR platform [50–52]. In general, AR systems fall into two groups. The first is lab-developed prototypes, while the second is commercial AR gadgets on the market. A broad outline of AR applications in recent studies can be found in Table 1. Cities and stakeholders can test out changes before implementing them in the real world using AR of real-world systems, places, or things based on DT.

**Table 1.** AR deployments in civil infrastructure studies.

| AR Application in Civil Infrastructure | | Current Progress | | Technical/Practical Challenges | References |
|---|---|---|---|---|---|
| Architecture | Tracking Project's Development | Discrepancy check | Prototype devices and software have been developed. | Outdoor environment Occlusion problem | [53] |
| | Quality control | Typical field implementations | | Accuracy requirement Occlusion problem | [54] |
| | Assisting workers | Collaborative Visualization | Theory | Communication stability | [55] |
| | | Telecommunication | Experiments | Communication stability Occlusion problem | [56] |
| | | Site safety | Experiments | Accuracy requirement Communication efficiency | [57] |
| BIM | | Multiple prototype devices and software have been developed | | Accuracy requirement Occlusion problem latency problem | [58,59] |
| Damage detection and SHM | | Multiple prototype devices and software have been developed | | Accuracy requirement Occlusion problem latency problem | [7,60] |
| Underground utilities | | Small-scale experiments | | Localization problem. Accuracy requirement Occlusion problem | [61,62] |
| Smart city | | Concepts | | Accuracy requirement Occlusion Problem Communication stability Privacy concern | [63] |

Based on a detailed review of the available literature, there is only one study about integrating AR technology with PMS. This study developed a mobile PMS on smartphones and a web-based PMS using Java programming language based on the location-based ser-

vices concept. Mobile PMS allows road users and engineers to report road defects. To perform maintenance activities, the field road engineers view the reports on the web-based PMS. In addition to developing AR technology, the mobile PMS allows field road engineers to obtain immediate reports, plan routes and resources for neighboring defects, and perform maintenance on-site [64].

Besides one study that proposes an improved local search heuristic using a social engineering optimizer as a metaheuristic [65] and another interesting modified discrete gravitational search algorithm which is also offered as a novel optimization algorithm [66], there is no significant research that integrates data collection, analysis, and visualization to produce a 3D visualized interactive result to move toward a smart city which is the most essential research gap. Tackling this severe gap, this study aims to collect the appropriate spatial data, apply GIS and PMS analysis, and produce the result as an AR experience utilizing the unity game engine.

## 3. Methodology

The three critical steps of the planned PMS are depicted in Figure 5. Each phase will be discussed in detail in the following paragraphs.

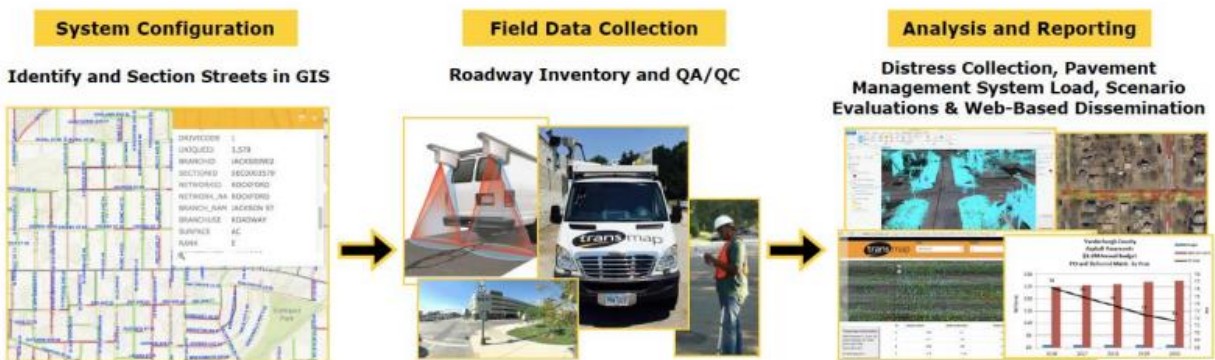

**Figure 5.** PMS process flow chart System configuration, reprinted with permission from [67].

The first step is creating a database with a GIS map and other pavement-related data. The designed PMS is based on a reliable database, and its reliability has a direct impact on future road status forecasting and M&R planning. Moreover, roadway data, traffic circumstances, pavement conditions, and all the essential expenses must be gathered for each street section. All obtained data is linked to the GIS map to establish the PMS GIS database. Creating an accurate GIS map takes a long time due to the amount of spatial analysis that must be evaluated and validated. Manual geocoding of information into a digital map may be required, as well as the collection and addition of additional roadway features.

Based on the requirements of the first phase, three-dimensional point clouds will be produced by the 3D LiDAR sensor covering the arterial roads of the study area. By minimizing the LiDAR field of view, the road surface can be obtained by removing data that is not relevant to the road surface. The areas will then be divided into frames on the road surface, each covering a section of the road.

### 3.1. Field Data Collection

The second phase entails the creation of road condition forecasts. Depending on the nature, number, and degree of the distress, the approach begins with identifying key condition indicators such as traffic loading data (vehicle/day), rainfall data, humidity data, etc. However, a pavement design approach should be established to collect the research area's critical attribute data. Pavement design's primary purpose is to build a low-cost road surface that meets site-specific efficiency, service life, regulatory criteria, and the city manager's financial goals.

One of the essential pavement design considerations is traffic loading, including road congestion, load factors and distribution, tire pressures, and suspension system characteristics. Rainfall, humidity in the pavement layers, temperature changes, and defrost durations are all factors to consider. In this paper, the data produced in a recent study are used [42], which is partially presented in Table 2. The available data are as follows:

- Name, start and endpoint, length, width, and functional class of the roads,
- Traffic data (vehicle/day),
- Primary diagnosis of causes of deterioration,
- Tests and actions made by the public work department and cost of work,
- Cost to users (annual cost).

**Table 2.** Sample of the data, reprinted with permission from [42].

| Name of the Street | Length (m) | Width (m) | Functional Class | Comfort | Estimated IRI (m/km) | Quality | Mesh Cracks | Shore Cracks | Longitudinal Cracks | Ripples/Ruts | Potholes |
|---|---|---|---|---|---|---|---|---|---|---|---|
| Major Street | 38 | 7 | Residential | High | 6.0 | 90% | Mild. Opening 5 mm or less | 10% | Mild. Opening 5 mm or less | —- | —- |
| Scott street | 282 | 6 | Residential | Medium | 8.0 | 60% | Mild. Opening 5 mm or less | 20% | Mild. Opening 5 mm or less | 40% | Mild. Opening 5 mm or less |

The state of the pavement is assessed using a low-cost action camera that is utilized in conjunction with the LiDAR sensor on top of the automobile to collect pavement data (Figure 6 shows the overview of the camera configuration and equipment applied). For the arterial roads of the study area, video is recorded at a maximum speed of 40 km/h. The camera is angled at a 40-degree angle toward the pavement. This design can collect a 4*3 square meter road section as a sample unit of pixels, meaning 1 pixel covers around 1 mm. Pavements in almost the exact location with similar features (for example, surface layer, traffic, and weather) should behave the same way. Estimating road conditions is commonly done by categorizing roadway sections into families and developing a predictive model for each pavement group. Two categories of pavement are available: the first ones are the road sections with cracks and the second ones are without cracks. However, considering the cracks will not cover the whole road assessment, the rest of the pavement evaluation factors are not considered in this paper due to diminishing the model complexity.

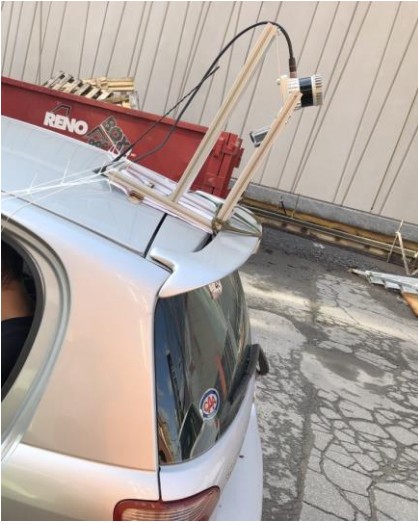

**Figure 6.** The data collection sensor is installed on top of the car, reprinted with permission from [68].

### 3.2. Analysis and Reporting

In the third phase, expenses, long-term objectives, emphasized specific components, and budget limits are used to build M&R plans. To begin, it is necessary to determine M&R solutions and unit pricing for various pavement kinds. The organization can choose a timeline that aligns with its operational plans. When setting priority standards, evaluating the pavement quality, distress type, pavement performance, volume of traffic, and project record is necessary. Figure 7 illustrates the standard approach to identifying management strategies for M&R on a road network based on how they are developed. Specific performance measures that need to be met depending on city agency decisions are available that may be discussed through various scenarios. It could be a percentage above a performance criterion or an average pavement performance score.

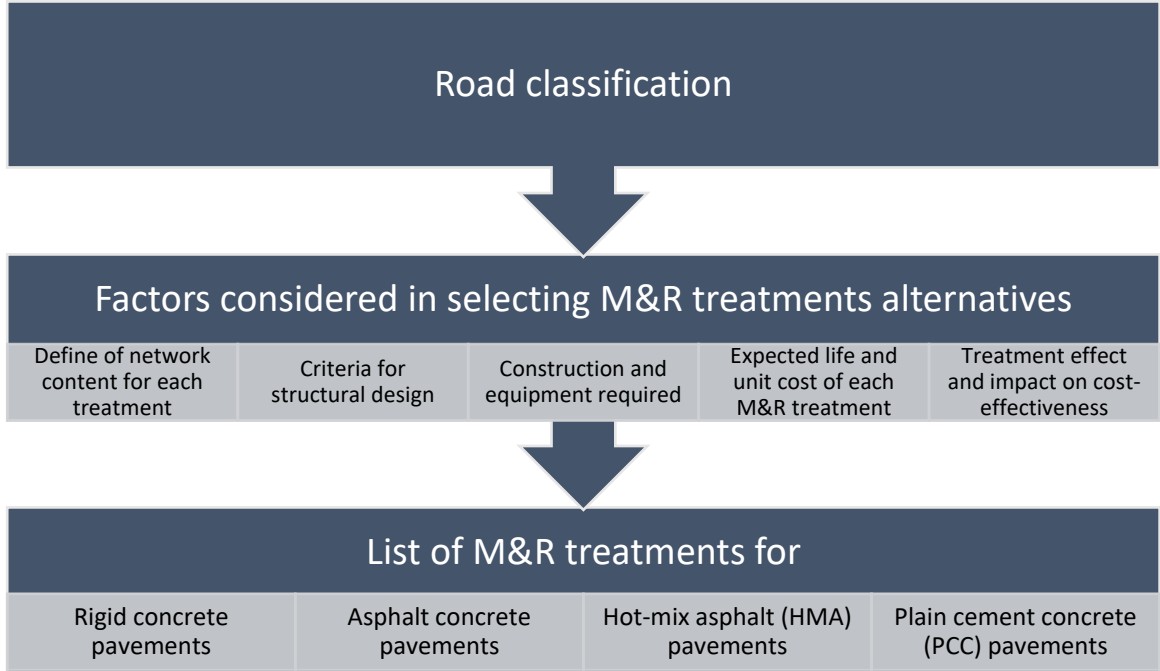

**Figure 7.** The process of developing practical M&R treatment strategies [69].

Having enough money to fix the road and satisfy those performance metrics will always be a difficulty, regardless of the objectives. The agency must determine the highest priority work projects and the optimal time to complete them to maximize the use of available funding. To provide the best visual output of the designed GIS-based PMS, it is necessary to display the results of the two primary phases in a 3D visualized platform.

However, providing the most understandable 3D city model to the stakeholders requires lots of data preparation and programming in a game engine. The produced model should also switch between different scenarios based on every stakeholder's priority. Depending on the time and the location of each city project, the role of stakeholders may vary a lot and impose different budget loads on the yearly planned budget for urban management.

Providing the optimum M&R treatment based on city stakeholders' priorities avoids wasting the financial resources of a city if it is presented and visualized in its best possible status. As a result, it is necessary to produce the CityEngine model before importing the model into the game engine because game engines are still not able to work with GIS data directly.

### 3.3. AR Output Visualization

On top of the AR output of our designed model, the CityEngine model is needed for studying urban areas. Modeling an entire city can be done semi-automatically with CityEngine, a procedural generation tool. Using algorithmic or rule-based methods in

procedural modeling allows the creation of large-scale scenes. It is possible to parametrize, control, or randomly set the rules that produce scenes and create and modify as many scenarios as needed [70].

Based on these capabilities, in this step of the methodology, first, the result of data analysis acquired by the combination of the previous three phases will enter the CityEngine, and then considering the priorities of the city officers and macro-management policies that will be the criterion in each executive project, the appropriate rules would be applied (the details are depicted in Figure 8). Formerly, for the visualization goal, a three-dimensional model will be prepared and imported to the Unity game engine of the designed AR experience.

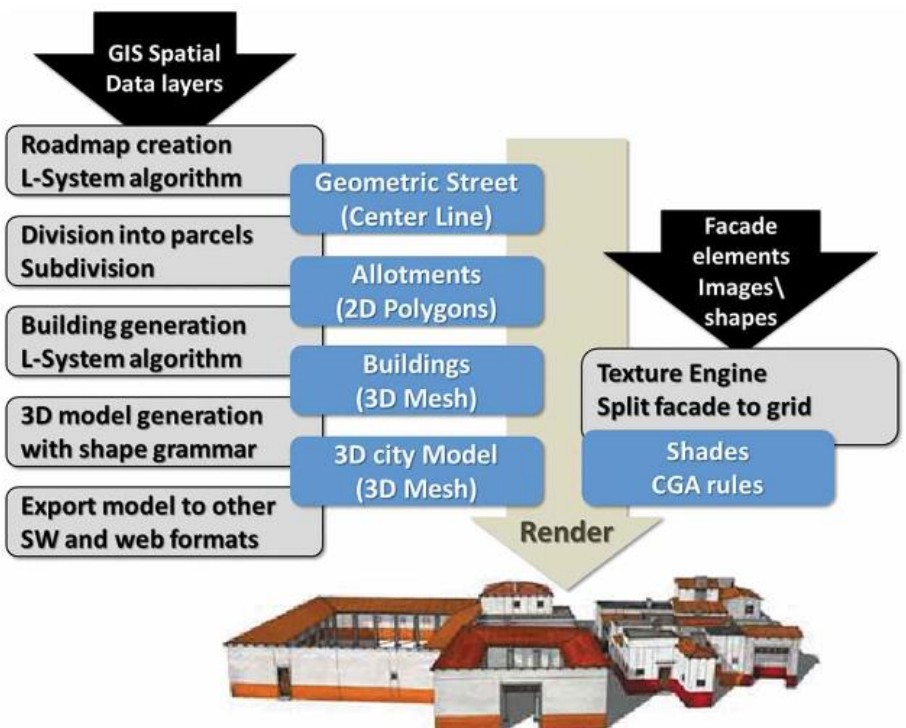

**Figure 8.** CityEngine step-by-step procedure [71].

This paper applies the Unity game engine to generate the necessary AR experience. Some characteristics of Unity are as follows:

- It has a complete game development environment,
- It runs on all major platforms and supports many target platforms (PC, mobile, console),
- It supports state-of-the-art 3D rendering.

For GIS and CAD data, Unity still has some restrictions. It works in a local coordinate system and cannot deal with georeferenced systems, which is the most unpleasant key factor. In a georeferenced system, a digital map or aerial photo's internal coordinate system can be related to a ground system of coordinates, and the most critical data entered the analysis model. Due to the importance of having access to the correct spatial georeferenced data, it must be converted (centered) before being imported into Unity. It should also be noted that the extent must be limited depending on the base unit used. As a result, formerly organized data becomes unstructured, making it difficult to engage.

Data preparation is needed to overcome the limitations of importing the CityEngine model into the Unity game engine. This step is a new term. In Unity, custom imports, scripts, and FBX models are used. FBX is a proprietary format for exchanging 3D geometry with Autodesk's freely available Software Development Kit (SDK). Access through the Unity game engine is now the leading exchange format for AR/VR applications and game

engines. The FBX model is the output of the CityEngine which will be imported here for the rest of the process. A data flow graph can be built visually and say what is needed to do with the data which has been imported. Figure 9 represents the view of Unity in the data preparation step.

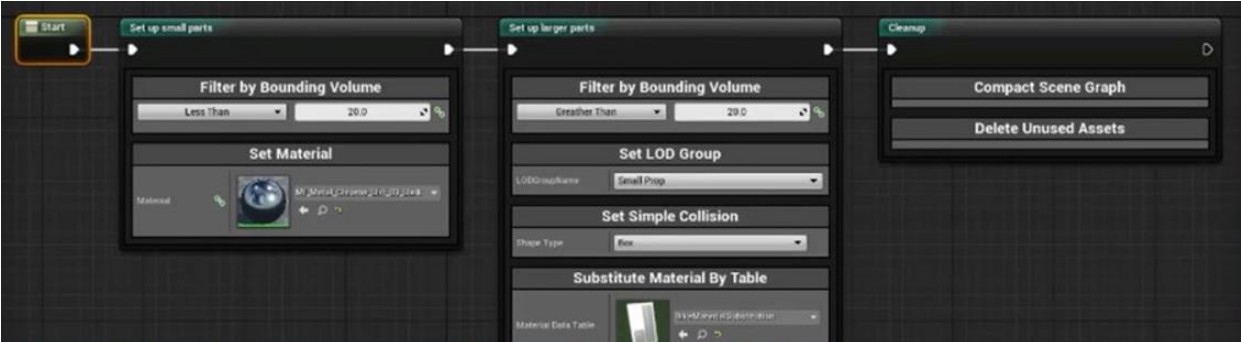

**Figure 9.** Unity data preparation view.

Until now, the only way to obtain geographic data in Unity was to use CityEngine, which has much connectivity for importing this type of data. For the methodology (depicted in Figure 10), it can be imported:

- BIM/CAD models using FBX data type,
- Base maps through the find map data mechanism,
- Synchronizing the data from ArcGIS Online and ArcGIS urban platforms.

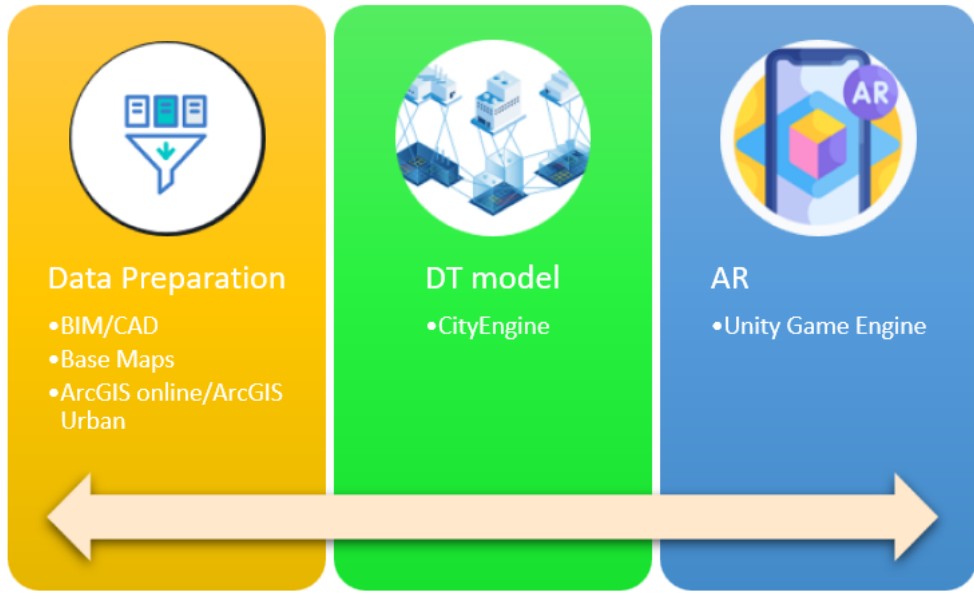

**Figure 10.** Data to information flow in the AR experience.

Building the scene in Unity, moving the camera to a place of interest in the study area, focusing on each road part, adding features to the desired road section, and replacing CityEngine preview materials with high-quality Unity texture material would be the final steps. This would result in a reliable visual and interactive 3D image of the road section and a real-time analysis of the proposed PMS's ideal scenario. The results will be presented in Section 4.

## 4. Results

Based on the methodology elaborated upon in Section 3, the rest of this section will provide a detailed step-by-step procedure of the implementation phase of this paper, including the study area presentation, data and model applied, and finally, the AR experience produced.

### 4.1. Study Area

The City of Châteauguay is a small to medium-sized city in the province of Québec (Canada), holding 50,815 inhabitants based on the 2021 census, with a road network of almost 500 km and a replacement value of around $1 billion. It is seeing rapid residential and industrial growth. Due to an outdated network, increased traffic, and numerous roads, pavement degradation occurs more quickly than anticipated. Like other Canadian municipalities, the challenge is that the annual budget is insufficient to maintain these continually deteriorating roads to an acceptable level, leading to a massive backlog of M&R work collecting or being delayed at a higher cost. Therefore, it is crucial to have a solid, dependable, and easy-to-implement PMS that is founded on the following features:

- Trustworthy data,
- Identifying the cause of the deterioration,
- Confirmation of this through relevant testing and the selection of actions that can address the deterioration's cause,
- A cost-benefit analysis of the life cycle,
- A visualized AR experience based on a real-time scenario to persuade city officers to invest in an interactive and intelligent PMS.

### 4.2. Data

The PMS is applied to the city of Châteauguay's pavements by gathering all essential data from the existing research [42,68], integrating it, and creating the most comprehensive GIS database. Street names, lane type, traffic data (vehicle/day), comfort index and estimated IRI, primary diagnosis of causes of deterioration for each road section, tests and actions taken by the public works department, cost of work, vehicle operation costs (VOC), and internal performance are all included in the attribute table of streets.

It would be necessary to extract pavement footprints before moving on to the primary 3D model of this analysis stage. Pavement polyline extraction is complex and requires more investigation. For pavement polyline extraction, first, it would be necessary to filter the LiDAR point cloud (LAS) dataset and access the road points. Then, the appropriate raster would be prepared to run the LAS point statistics as a raster tool. Applying another tool to convert this raster to a polyline makes us move forward with the extraction of pavement polylines. However, some graphical works are needed to clean the feature class created and access the best appropriate geometrical shape of the pavements. Figure 11 provides a schematic view of all the steps required in this study before entering the CityEngine model.

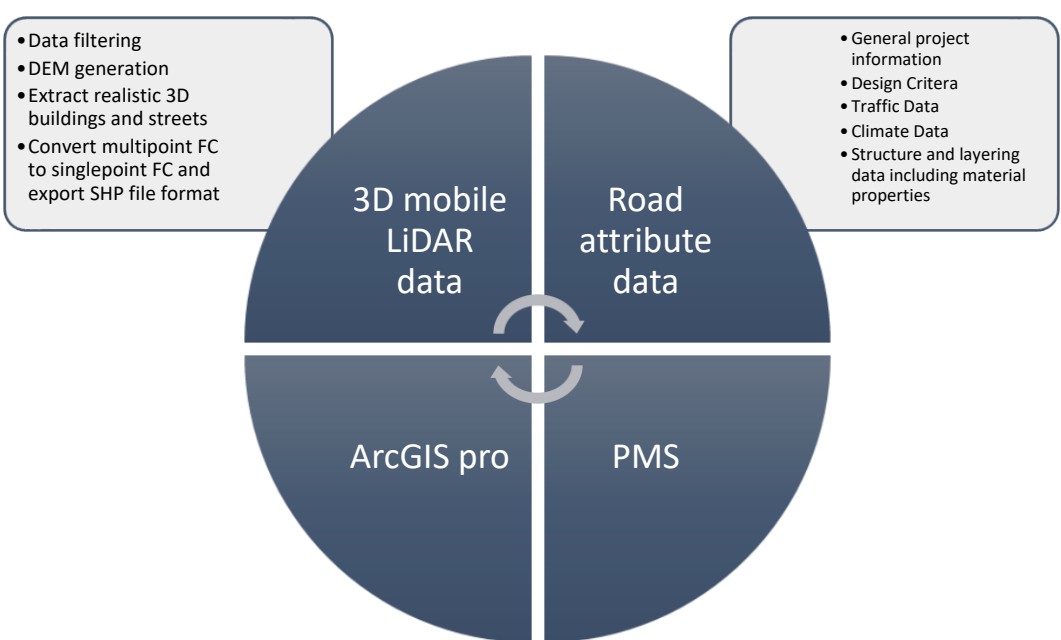

**Figure 11.** Different steps of the analysis run on the data to make the designed approach.

### 4.3. Creating the CityEngine Model

In this study, streets were imported as Shapefiles. Polylines that were put into City-Engine as street segments comprise the data. The street network layer was generated automatically, and the Viewport showed street centerlines. Using the "Resolve Conflicting Shapes" tools in Cleanup Graph, various graphical disorders in the street centerlines shapefile are automatically corrected.

In this CityEngine configuration, the Computer Graphics and Applications (CGA) syntax defines streets. A CGA rule file is a collection of rules that define how geometry is created. A CGA rule file is usually allocated to a shape. The street model in this project was created using CGA from the Esri library. The Essential Street CGA file includes screen resolution, road network, side configuration, parking design, bike lanes, landscaping, street items, and population.

It should be noted that because the GIS database includes all the different attributes of a single road section, the created city model encompasses those data. Figure 12 shows that all the data is presented by clicking on every part of the city's pavements.

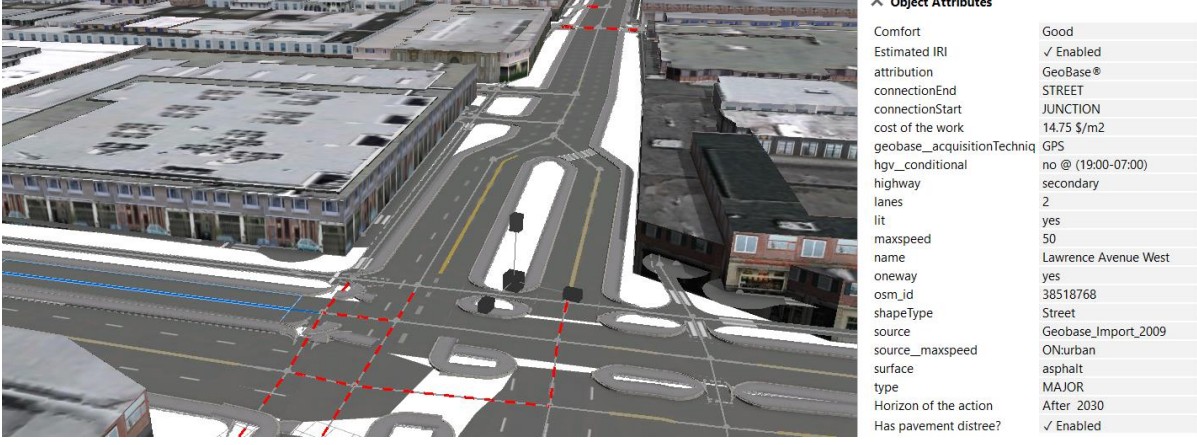

**Figure 12.** CityEngine model of the designed PMS.

Later, to apply the generated model in the Unity game engine, it is necessary to export the model as an Autodesk FBX model. Figure 13 shows the details.

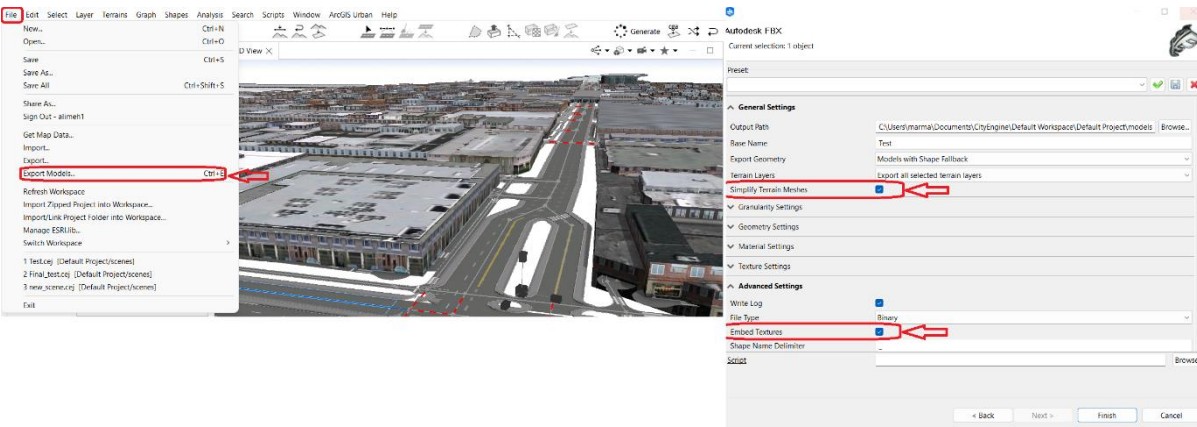

**Figure 13.** Details of the Autodesk FBX model settings.

### 4.4. AR Experience

After dragging and dropping the FBX model into Unity, the camera must be moved to a study area point of interest. It would only be the beginning of developing the AR experience, including adding AR extensions via AR camera motion controllers and a slew of other features dependent on research requirements. It would then be feasible to play around with the created AR experience from the point when the camera is positioned on it before adding the features by pressing the "Play" button on top of the screen (Figure 14 shows how it can be played to move around the built AR experience). Having access to the AR experience developed here can optimize the current PMS in three distinct aspects:

- Optimize for the initial cost of the PMS
- Optimize for (cost-neutral) pavement performance
- Optimize for the life cycle cost of the PMS

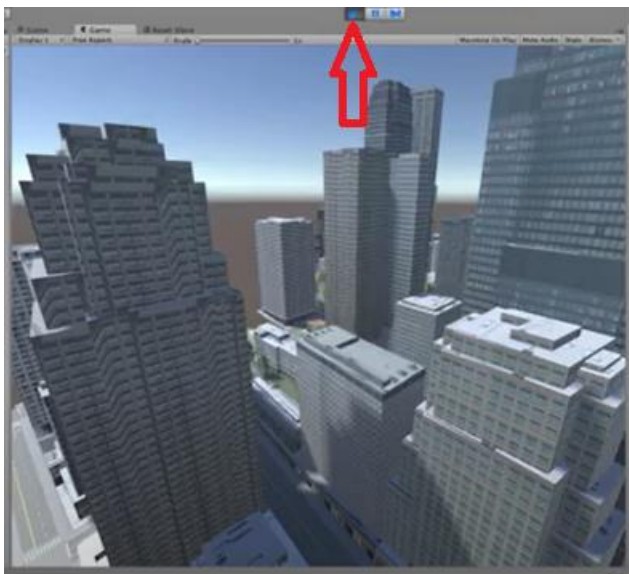

**Figure 14.** Playing the AR experience built based on the designed PMS by hitting the button on top of the window.

### 4.5. Validation

In this paper, the triangulation method was used to validate the study approach because of the integration of different technologies applied to produce the final 3D AR experience of the study region. Using triangulation can cross-verify methodologies from more than two sources to validate it. Using a variety of instruments allows us to assess the consistency of the results and identify any factors that might influence them adversely.

By combining several techniques or sources of data, triangulation is utilized to create a more thorough knowledge of events [72]. There are four different types of triangulation, according to two excellent recent pieces of research: data, investigator, theory, and methodology triangulation [72,73].

It is common to practice employing three separate data sources to boost a study's reliability. The expert interviews' responses would be reviewed throughout the analysis to identify areas of agreement and disagreement. Categorizing each stakeholder group for the program, we are evaluating is a crucial strategy. It also ensures that an equal number of stakeholders represents each stakeholder group.

Here in this paper, we applied data triangulation. To acquire the opinion of the desired stakeholders, we asked a group of urban municipal experts and citizens in the study area to participate in the validation process. Tables 3 and 4 are two sample questionnaires filled out to endorse the approach. It should be mentioned that before filling out the questionnaire, the system was presented in detail to them.

**Table 3.** The first sample of the validation Questionnaire.

|  | Low | Medium | High |
|---|---|---|---|
| What was the quality of the implementation approach? | ☐ | ☐ | ☐ |
| To what extent were the program objectives met? | ☐ | ☐ | ☐ |

**Table 4.** The second sample of the validation Questionnaire.

| Question | Answer |
|---|---|
| What other impacts did this approach have? | |
| How could the approach be improved? | |

The validity is established based on the results acquired in this step. Due to the different stakeholder groups involved in the approach, this type of triangulation is probably the most popular and efficient one in the integrated framework containing different technologies (The framework of the triangulation validation method for this paper is displayed in Figure 15).

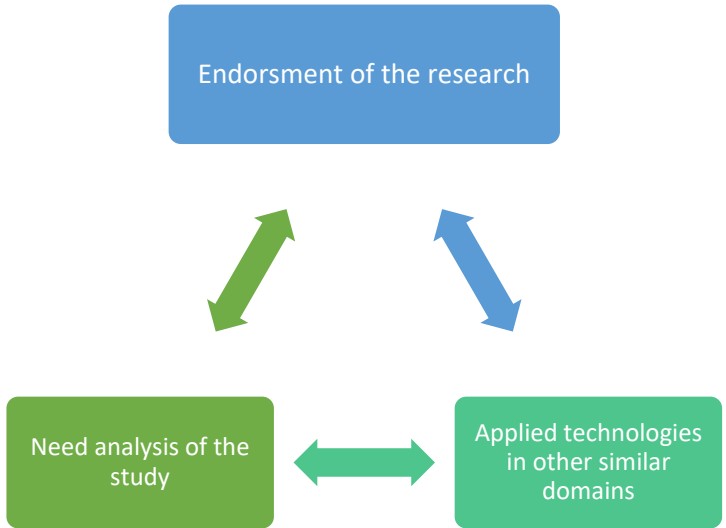

**Figure 15.** Triangulation schema.

## 5. Conclusions and Future Trends

Considering the context of a smart city and its related urban infrastructure management system, this paper concludes that making a comprehensive smart PMS and applying the recently emerged cutting edged technologies to overcome the previous issues is a necessity in the intelligent city concept among researchers. This paper reports on innovative developments in infrastructure monitoring that can help create intelligent pavement maintenance in urban areas and save money for taxpayers. To do this, the process begins by determining why the PMS requires using contemporary innovative technologies and AI-based platforms. The PMS was then divided into three main phases. Due to the fact that the significance of comprehensive GIS data and analysis in urban infrastructure studies is particularly essential, the research technique combines 3D LiDAR data and attribute data from the city of Châteauguay's pavement network to obtain the most reliable data. Since creating the AR experience in urban infrastructure research plays a colorful role nowadays, and almost no research has been found focusing on this framework, this paper concentrated on building the CityEngine model of the study area on top of the designed and implemented AR experience.

The results of the designed approach and the verification of the framework by two groups of stakeholders satisfy the paper's primary objective. By providing stakeholders with DTs of the city pavements before the implementation of each PMS, the final visualized output can prevent millions of dollars in waste every year. This paper concludes that although linking spatial data and game engines is still impossible without middleware, the future of brilliant PMS and smart cities is unimaginable by neglecting the role of 3D visualization platforms, including AR environments.

Intelligent data acquisition and assessment approaches for functional and structural evaluations may help future DSS for road maintenance. Compared with conventional systems and research methodologies, this integrated strategy could save money and resources while improving the efficacy of infrastructure institutions' maintenance and monitoring programs. It must derive insights at multiple levels, both on the surface and within the pavement, and identify and forecast all forms of structural and functional faults and flaws as part of a coordinated DSS. They will also require close collaboration between civil and IT specialists to establish a comprehensive framework that encompasses all smart technologies in their DSS.

The AR experience is complicated by the convergence of GIS and AR technology. Due to the current state of development of AR geolocation systems, the positioning of the virtual objects may not be correct. The system could place one infrastructure in the wrong place or even next to another. These issues lead to a poor user experience while using

AR environments. As a signal of the limitations of this paper, it should be emphasized that working with geospatial data is not supported inside the game engines as of now. It means all the spatial analysis should be done before entering the AR environment. These limitations make converting the data into 3D information inside the AR experience more time-consuming and, as a result, more expensive. By overcoming the stated limit, there would be no need to receive the help of third-party platforms such as CityEngine as an interface between GIS and AR.

**Author Contributions:** Conceptualization, methodology and software, writing and original draft preparation, visualization and investigation, M.M.; supervision, review and editing, G.J.A. All authors have read and agreed to the published version of the manuscript.

**Funding:** This research received no external funding.

**Institutional Review Board Statement:** Not applicable.

**Informed Consent Statement:** Not applicable.

**Data Availability Statement:** Not applicable.

**Conflicts of Interest:** The authors declare no conflict of interest.

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
