# Peer review of "Building an Augmented Reality Experience on Top of a Smart Pavement Management System"

_buildings, doi:10.3390/buildings12111915_

Round 1
Reviewer 1 Report
1. the format of the designed tables can be improved, such as tables 3 and 4.
2. it is suggested to combine sections 2 and 3 since this article is not a literature review article and section 3 puts too much information on the literature summary, especially some summaries are not highly relevant to the research topic of the manuscript.
3. the modelling construction try to simulate both roads and city buildings well, actually in the PMS people are more interested in road condition itself, and the involvement of buildings may need extra energy .
Author Response
Dear Reviewer 1,
We thank the editor and the reviewers for their valuable time reviewing our manuscript and providing helpful comments to help improve our proposal.
The authors have carefully considered your comments and addressed each of them. While we hope the revised manuscript meets your standards, we welcome further constructive comments. The following is a point-by-point response to each comment provided, and modifications are done using MS-Word track changes.
Sincerely yours,

Reviewer 2 Report
Overall, the work is organized well and the logic looks nice. However, there are some points that should be addressed in order to improve its presentation and scientific content:
1. The abstract of the paper does not highlight the key points and is illogical.
2. The introduction lacks a motivation to undertake this research. To this reviewer, the problem has to be properly defined showing the right level of motivation at both scientific and practical domains.
3. This manuscript fails to support the novelty and the contribution of the work done.
4. Literature review is poorly developed. Although there is a reasonable number of references, the actual review says little in terms of whether they fail or succeed to address the underlying problem(s) and how your research fits in. In addition, compared with the relevant research, the innovation of this paper should be highlighted. For example,
Recycling of spent Lithium-ion Batteries: A comprehensive review for identification of main challenges and future research trends, Sustainable Energy Technologies and Assessments, 2022, 53,102447.
An Enhanced Social Engineering Optimizer for Solving an Energy-Efficient Disassembly Line Balancing Problem Based on Bucket Brigades and Cloud Theory. IEEE Transactions on Industrial Informatics, 2022.
Multi-objective scheduling of priority-based rescue vehicles to extinguish forest fires using a multi-objective discrete gravitational search algorithm, Information Sciences, 2022, 608:578-596.
5. Discuss the practical implications. In addition, future work needs to explained more in detail.
6. The paper needs editing for language and grammar. I would recommend English language proof reading.
Author Response
Dear Reviewer 2,
We thank the editor and the reviewers for their valuable time reviewing our manuscript and providing helpful comments to help improve our proposal.
The authors have carefully considered your comments and addressed each of them. While we hope the revised manuscript meets your standards, we welcome further constructive comments. The following is a point-by-point response to each comment provided, and modifications are done using MS-Word track changes.
Sincerely yours,

Reviewer 3 Report
The manuscript proposes an AR experience on top of a smart pavement management system by applying CityEngine capabilities to the aim of provide more efficient, less costly, safer, and real-time procedures.
The manuscript is well structured and there is a logical flow among the various section. Method is well described. I suggest improving the section of results better clarifying how this approach allows optimizing the PMS.
Author Response
Dear Reviewer 3,
We thank the editor and the reviewers for their valuable time reviewing our manuscript and providing useful comments to help improve our proposal.
The authors have carefully considered your comments and addressed each of them. While we hope the revised manuscript meets your standards, we welcome further constructive comments. The following is a point-by-point response to each comment provided, and modifications are done using MS-Word track changes.
Sincerely yours,

Round 2
Reviewer 2 Report
I consider that this revised manuscript may be acceptable for publication.